# Chromatin-prebound Crm1 recruits Nup98-HoxA9 fusion to induce aberrant expression of *Hox* cluster genes

**Masahiro Oka[1,2]\*, Sonoko Mura[3], Kohji Yamada[1], Percival Sangel[1], Saki Hirata[4], Kazumitsu Maehara[4], Koichi Kawakami[5], Taro Tachibana[6], Yasuyuki Ohkawa[4], Hiroshi Kimura[7], Yoshihiro Yoneda[2,8]\***

[1]Laboratory of Nuclear Transport Dynamics, National Institutes of Biomedical Innovation, Health and Nutrition, Osaka, Japan; [2]Laboratory of Biomedical Innovation, Graduate School of Pharmaceutical Sciences, Osaka University, Osaka, Japan; [3]Graduate School of Frontier Biosciences, Osaka University, Osaka, Japan; [4]Department of Advanced Medical Initiatives, Kyushu University, Fukuoka, Japan; [5]Division of Molecular and Developmental Biology, National Institute of Genetics, Shizuoka, Japan; [6]Department of Bioengineering, Osaka City University, Graduate School of Engineering, Osaka, Japan; [7]Department of Biological Sciences, Graduate School of Bioscience and Technology, Tokyo Institute of Technology, Yokohama, Japan; [8]National Institutes of Biomedical Innovation, National Institutes of Biomedical Innovation, Health and Nutrition, Osaka, Japan

**Abstract** The nucleoporin Nup98 is frequently rearranged to form leukemogenic Nup98-fusion proteins with various partners. However, their function remains largely elusive. Here, we show that Nup98-HoxA9, a fusion between Nup98 and the homeobox transcription factor HoxA9, forms nuclear aggregates that frequently associate with facultative heterochromatin. We demonstrate that stable expression of Nup98-HoxA9 in mouse embryonic stem cells selectively induces the expression of *Hox* cluster genes. Genome-wide binding site analysis revealed that Nup98-HoxA9 is preferentially targeted and accumulated at *Hox* cluster regions where the export factor Crm1 is originally prebound. In addition, leptomycin B, an inhibitor of Crm1, disassembled nuclear Nup98-HoxA9 dots, resulting in the loss of chromatin binding of Nup98-HoxA9 and Nup98-HoxA9-mediated activation of *Hox* genes. Collectively, our results indicate that highly selective targeting of Nup98-fusion proteins to *Hox* cluster regions via prebound Crm1 induces the formation of higher order chromatin structures that causes aberrant *Hox* gene regulation.

\*For correspondence: moka@ nibiohn.go.jp (MO); y-yoneda@ nibiohn.go.jp (YY)

**Competing interests:** The authors declare that no competing interests exist.

## Introduction

The nucleoporin Nup98 is a mobile component of the nuclear pore complex (NPC) (*Griffis et al., 2002*; *Rabut et al., 2004*; *Oka et al., 2010*), a sole gateway for selective nucleocytoplasmic macro-molecular traffic. Nup98 is essential for such fundamental functions of NPC as selective nucleocyto-plasmic transport (*Radu et al., 1995*; *Powers et al., 1997*; *Zolotukhin and Felber, 1999*; *Oka et al., 2010*) and maintenance of the permeability barrier (*Hulsmann et al., 2012*; *Laurell et al., 2011*). Besides, Nup98 is known as a multifunctional nucleoporin; it has been shown that Nup98 is involved in gene regulation (*Capelson et al., 2010*; *Kalverda et al., 2010*; *Liang et al., 2013*; *Light et al., 2013*), posttranscriptional regulation of specific sets of messenger RNAs (mRNAs) (*Singer et al., 2012*), mitotic spindle assembly (*Cross and Powers,*

**eLife digest** The nucleus of a eukaryotic cell (which includes plant and animal cells) contains most of the cell's genetic material in the form of carefully packaged strands of DNA. Genes are stretches of DNA that contain the instructions needed to produce the proteins and RNA molecules that the cell needs to survive. These molecules move across the membrane that surrounds the nucleus through pores made of proteins. One of these pore-forming proteins is called Nup98. The gene that produces Nup98 is frequently mutated in leukemia, where part of it becomes fused to regions of other unrelated genes. The proteins made from these combined genes are known as "fusion proteins".

The Nup98-HoxA9 fusion protein has been well studied, and appears to cause leukemia by interfering with the process called ("cell differentiation") by which stem cells specialize to form different types of blood cells. During cell differentiation, cells change which sets of genes they activate to become specific types of cells. A family of genes called *Hox* genes (to which the gene for HoxA9 belongs) is critical in cell differentiation and thus must be fine-tuned. It is also known that the *Hox* genes form clusters, and its activation is partly controlled by how tightly the DNA is packaged.

Previous studies have shown that the Nup98-HoxA9 fusion protein takes on the form of small dots in the nucleus. Oka et al. have now tracked how these proteins are distributed inside of the nucleus, and examined which part of the DNA they bind to, in more detail. This revealed that the dots of Nup98-HoxA9 tend to associate with tightly packed DNA, especially on *Hox* cluster genes, and activate these genes.

Oka et al. further found that a protein called Crm1, which is well known as a nuclear export factor that carries molecules out of the nucleus through the pore, is already bound to the *Hox* cluster genes in the nucleus and recruits the Nup98-HoxA9 protein. This interaction may change how the *Hox* gene is packaged in the nucleus. A future challenge will be to reveal how the Nup98-HoxA9 fusion protein and Crm1 on *Hox* cluster genes control gene expression.

2011), mitotic checkpoint (*Jeganathan et al., 2005*; *Salsi et al., 2014*), and NPC disassembly (*Laurell et al., 2011*).

In leukemia, Nup98 is frequently found in the form of Nup98-fusions, which consist of N-terminal half of Nup98 containing multiple phenylalanine-glycine (FG) repeats and C-terminus of various partner proteins (*Gough et al., 2011*). More than 30 different proteins with various physiological functions have been reported as Nup98 fusion partners (reviewed in (*Gough et al., 2011*)). However, the molecular mechanism of Nup98-fusion mediated leukemogenesis is still largely unknown.

Nup98-HoxA9 is one of the most frequent Nup98-fusion resulting from t(7;11)(p15;p15) chromosomal translocation associated with acute myeloid leukemia, myelodysplastic syndrome, and chronic myeloid leukemia (*Nakamura et al., 1996*; *Borrow et al., 1996*; *Nishiyama et al., 1999*; *Yamamoto et al., 2000*). Indeed, the ectopic expression of Nup98-HoxA9 induces leukemia in mice (*Kroon et al., 2001*; *Iwasaki et al., 2005*; *Dash et al., 2002*). It also has been shown that Nup98-HoxA9 inhibits hematopoietic cell differentiation (*Kroon et al., 2001*; *Calvo et al., 2002*; *Takeda et al., 2006*; *Chung et al., 2006*; *Yassin et al., 2009*) and enhances symmetric division of hematopoietic precursor cells in vitro(*Wu et al., 2007*), suggesting that Nup98-HoxA9 contributes to leukemogenesis most likely by impairing cellular differentiation.

With regard to its molecular function, Nup98-HoxA9 was shown to act as a transcriptional regulator (*Kasper et al., 1999*; *Ghannam et al., 2004*; *Bei et al., 2005*; *Yassin et al., 2009*). In addition, genome-wide gene expression analysis revealed that ectopic expression of Nup98-HoxA9 causes upregulation, rather than downregulation, of numerous genes (*Ghannam et al., 2004*; *Takeda et al., 2006*). Mechanistically, the FG repeat of Nup98 is known to associate with CREB-binding protein (CBP)/p300 (*Kasper et al., 1999*), a histone acetyltransferase that functions as a transcriptional co-activator, and histone deacetylase (HDAC) 1 (*Bai et al., 2006*). However, the interaction of Nup98 FG repeats with p300 by itself is not sufficient to promote self-renewal of hematopoietic stem cells (*Yung et al., 2011*). Thus, the exact function of Nup98-HoxA9 still remains unclear.

Here, we demonstrate that Nup98-HoxA9 is specifically recruited to the vicinity of *Hox* cluster genes via chromosomally bound Crm1, a nuclear export factor that was originally identified as a protein required for maintaining the chromosomal structure in yeast (*Adachi and Yanagida, 1989*), to activate *Hox* cluster regions.

## Results

### Nup98-HoxA9 dots associate with facultative heterochromatin

Previous reports have demonstrated that Nup98-HoxA9 fusion protein localizes as distinct small dots in the nucleus (*Kasper et al., 1999*; *Bai et al., 2006*; *Xu and Powers, 2010*; *Oka et al., 2010*). To address the role of Nup98-HoxA9, we first characterized in detail the intranuclear distribution of these dots. When expressed in HeLa cells, the enhanced green fluorescent protein (EGFP)-tagged Nup98-HoxA9 was easily distinguished as small nuclear dots. In contrast, Nup98 FG repeats were detected as large dots mostly associated with the nucleolus, and HoxA9 C-terminus containing the homeodomain, HoxA9-Ct, showed an even nucleopolasmic distribution (*Figure 1A*), consistent with earlier reports (*Oka et al., 2010*; *Xu and Powers, 2010*). We noticed that Nup98-HoxA9 dots were not randomly localized within the nucleus, but were frequently adjacent to 4',6-diamidino-2-phenylindole (DAPI)-dense heterochromatin regions (*Figure 1B*, *Figure 1—figure supplement 1*) as revealed by co-staining with DAPI. Furthermore, immunocytochemical analysis using monoclonal antibodies specific to histone modifications (*Kimura et al., 2008*) (*Figure 1C*) revealed that those dots were frequently associated with H3K9me2 and H3K27me3, facultative heterochromatin markers (*Trojer and Reinberg, 2007*), but not with the marker for constitutive heterochromatin H3K9me3 (*Peters et al., 2003*, *Rice et al., 2003*). These results indicate that Nup98-HoxA9 dots are not randomly positioned in the nucleus but are associated with specific chromatin domains.

### Nup98-HoxA9 dots show a cell-type specific localization pattern

We hypothesized that if Nup98-HoxA9 associates with specific chromatin modifications, its intranuclear distribution pattern may differ depending on the cell type. Therefore, we next expressed Nup98-HoxA9 in mouse embryonic stem (ES) cells, whose chromatin modifications are quite different from other cell types (*Meshorer and Misteli, 2006*). We found that although Nup98-HoxA9 nuclear dots could also be observed in ES cells, they were fewer in number and more heterogeneous in size compared with dots revealed in HeLa cells or NIH 3T3 cells (*Figure 2A*). Immunocytochemical analysis in ES cells (*Figure 2—figure supplement 1*) also showed the partial co-localization of Nup98-HoxA9 dots with H3K9me2 and H3K27me3, but less with H3K9me3. These findings indicated that the intranuclear distribution of Nup98-HoxA9 indeed changes depending on the cell type, although we cannot exclude the possibility that the variation in the transgene expression level caused these differences. Together, these results suggested that Nup98-HoxA9 might be involved in a cell type-specific gene and/or chromatin regulation.

### Nup98-HoxA9 expressing ES cells show flattened morphology and resistance to spontaneous differentiation

To gain more insights into the function of Nup98-HoxA9, which most likely associates with the impairment of cell differentiation (see Introduction), we tried to obtain stable cell lines expressing the FLAG-Nup98-HoxA9 fusion in ES cells using transposon vectors (*Kawakami and Noda, 2004*; *Urasaki et al., 2006*; *Oka et al., 2013*). Immunocytochemical analysis using stable ES clones demonstrated that FLAG-Nup98-HoxA9 had a similar localization pattern to transiently expressed EGFP-Nup98-HoxA9 in ES cells (*Figure 2B*). Immunoblotting using monoclonal antibody that is raised against the N-terminal (A.A. 1–466) of Nup98 (*Fukuhara et al., 2005*) revealed that FLAG-Nup98-HoxA9 was not overexpressed compared with endogenous Nup98 (*Figure 2—figure supplement 2*). Unexpectedly, we found that FLAG-Nup98-HoxA9 expressing stable ES cell clones (Nup98-HoxA9 ES) had distinct flat colonies, which was in contrast to tighter colonies of spherical shape detected in parental ES, Nup98FG, or HoxA9-Ct expressing ES cells (*Figure 2C*). Further experiments revealed that Nup98-HoxA9 ES cells expressed stem cell marker genes, and had proliferation rates comparable to those of parental ES cells, indicating that they were not differentiated (data not shown). In addition, Nup98-HoxA9 ES cells show more resistance to spontaneous cell differentiation

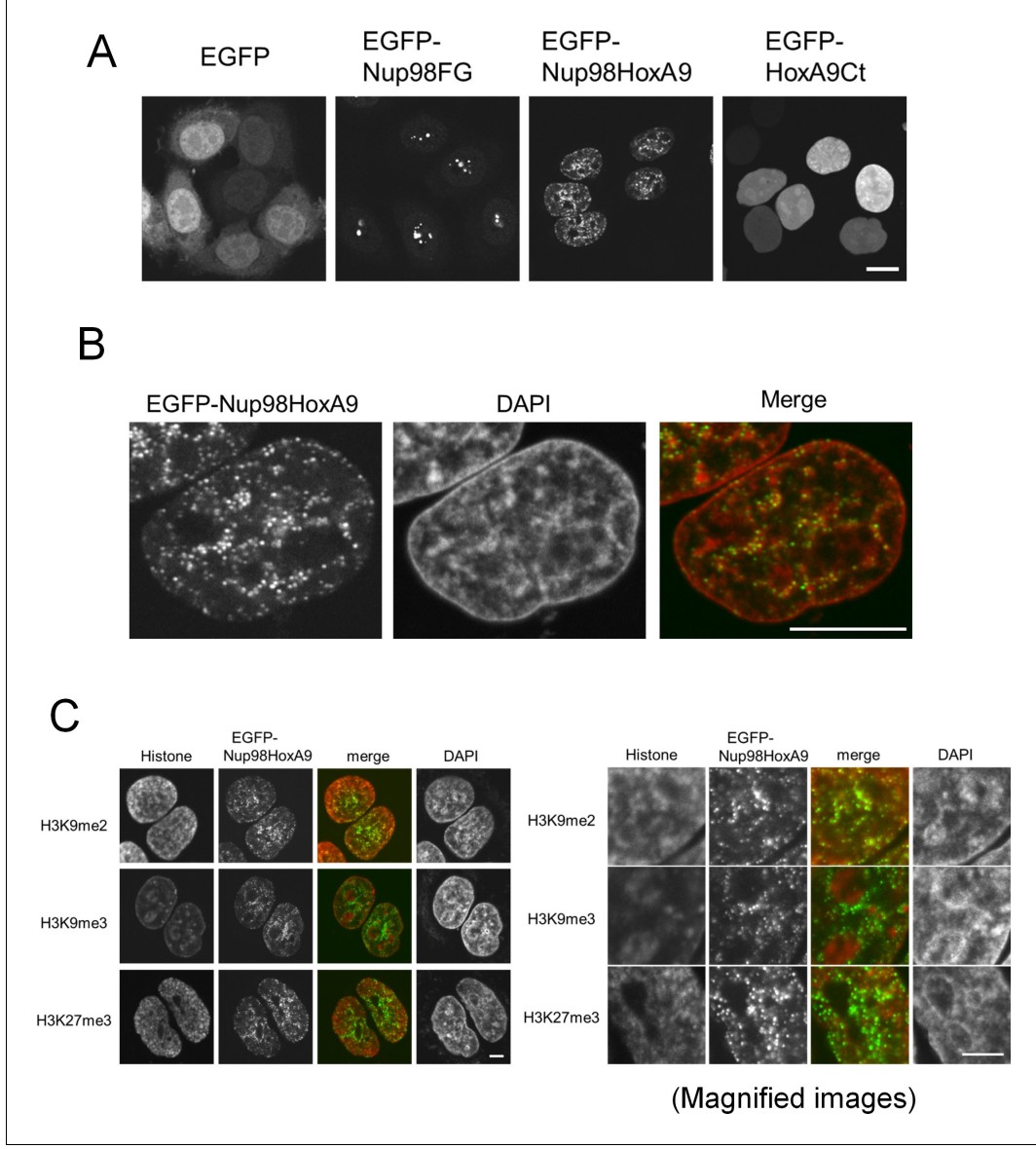

**Figure 1.** Nup98-HoxA9 dots associate with facultative heterochromatin. (**A**) Subcellular localization of Nup98-HoxA9 and its truncated mutants. EGFP, EGFP-Nup98FG, EGFP-Nup98-HoxA9, or EGFP-HoxA9-Ct expressing plasmids were transfected into HeLa cells for 24 hr and observed. Bar, 10 μm. (**B**) Confocal microscopy analysis of EGFP-Nup98-HoxA9 in HeLa cells. A merged image shows EGFP-Nup98-HoxA9 (green) and DAPI (red). Bar, 10 μm. (**C**) Association of Nup98-HoxA9 dots with specific histone modifications. HeLa cells were transfected with the EGFP-Nup98-HoxA9 expressing plasmid. Twenty-four hours after transfection, cells were fixed and stained with antibodies against indicated histone modifications. DAPI staining was used to visualize the nuclei. Bar, 5 μm. DAPI, 4′,6-diamidino-2-phenylindole; EGFP, enhanced green fluorescent protein.

The following figure supplement is available for figure 1:

**Figure supplement 1.** Serial z-sectioning of EGFP-Nup98-HoxA9 in HeLa cell.

in the absence of leukemia inhibitory factor (LIF), compared with parental ES, Nup98FG, or HoxA9-Ct expressing ES clones (*Figure 2D* and *Figure 2—figure supplement 3*).

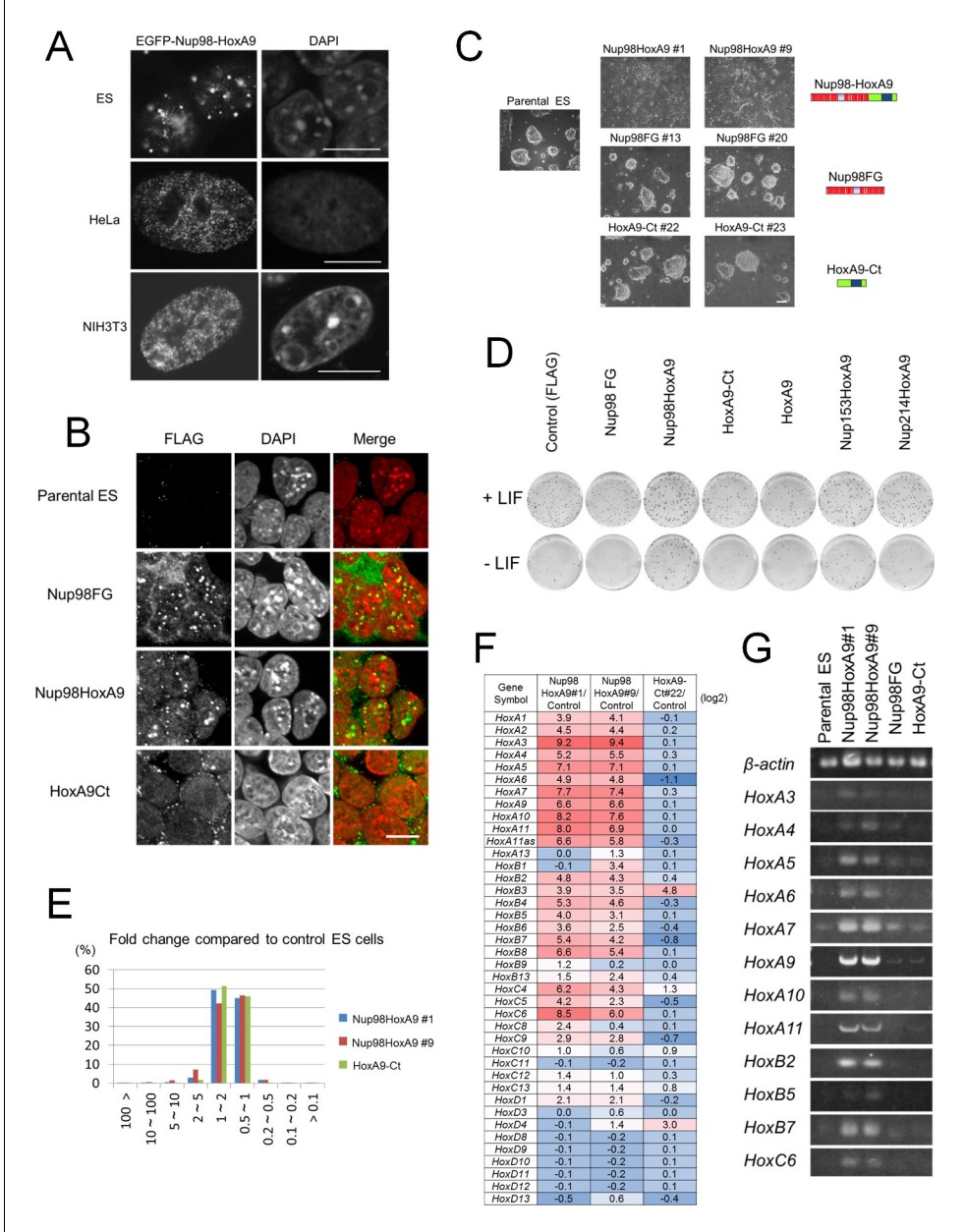

**Figure 2.** Nup98-HoxA9 evokes the expression of *Hox* cluster genes in ES cells. (**A**) Differential intranuclear localization of Nup98-HoxA9 in mouse ES, HeLa and NIH3T3 cells. Cells were transfected with the EGFP-Nup98-HoxA9 expressing plasmid for 24 hr, fixed, and stained with DAPI. Samples were analyzed using confocal microscopy. Bar, 10 μm. (**B**) Subcellular localization of Nup98-HoxA9 and its truncated mutants in ES cells. ES cell clones expressing FLAG-tagged Nup98FG, Nup98-HoxA9, or HoxA9-Ct were fixed and stained with an anti-FLAG (polyclonal) antibody. Nuclei were stained with DAPI. Merged images show FLAG staining (green) and DAPI (red). Bar, 10 μm. (**C**) Cell morphology of stable ES cell lines expressing FLAG-tagged Nup98-HoxA9, Nup98FG, or HoxA9-Ct. Bar, 100 μm. (**D**) Differentiation assay of stable ES cell clones. ES cells stably expressing FLAG (control), FLAG-tagged Nup98FG, Nup98-HoxA9, HoxA9-Ct, HoxA9, Nup153-HoxA9, or Nup214-HoxA9 were plated at a density of $10^3$ cells per well in 12-well plates either in the presence or absence of LIF. After 5 d, the plates were fixed and stained with alkaline phosphatase, a marker for undifferentiated stem cells. (**E**) Gene expression profiling of ES cell lines stably expressing FLAG-tagged Nup98-HoxA9 (#1 and #9; two independent clones) and HoxA9-Ct is compared with that of parental ES cells. A greater than 10-fold upregulation of gene expression was commonly observed in both Nup98-HoxA9 clone #1 and Nup98-HoxA9 clone #9 cells. (**F**) Hox gene expression profiling. The log2 fold ratios of the normalized signal value of *Hox* cluster genes from Nup98-HoxA9 or HoxA9-Ct expressing ES cells relative to signals from parental control ES cells are indicated. (**G**) Upregulation of *Hox* cluster genes in FLAG-Nup98-HoxA9 expressing ES cell lines was confirmed using semi-quantitative polymerase chain reaction. DAPI, 4',6-diamidino-2-phenylindole; EGFP, enhanced green fluorescent protein; ES, embryonic stem; LIF, leukemia inhibitory factor

The following figure supplements are available for figure 2:

*Figure 2. continued on next page*

*Figure 2. Continued*

**Figure supplement 1.** Co-staining of Nup98-HoxA9 with various histone marks.
**Figure supplement 2.** Protein expression levels of Nup98-HoxA9 and Nup98FG.
**Figure supplement 3.** Differentiation assay of stable embryonic stem cell clones.

## Nup98-HoxA9 expression in ES cells induces expression of *Hox* cluster genes

Next, to elucidate whether Nup98-HoxA9 affected global gene expression, we performed DNA microarray analysis using the following four clones: two independent Nup98-HoxA9 ES clones (clone #1 and clone #9), parental ES cells, and the HoxA9-Ct ES clone. We found that a small fraction of genes (less than 1%) were strongly induced (more than 10-fold) in Nup98-HoxA9 ES cells compared with their expression levels in parental or HoxA9-Ct ES cells (*Figure 2E*). Interestingly, we noticed that *Hox* cluster genes were highly activated in Nup98-HoxA9 ES cells (*Figure 2F*). In particular, more than a half of the top 32 upregulated genes, which were common in both of Nup98-HoxA9 clone #1 and Nup98-HoxA9 clone #9, were *Hox* cluster genes belonging to *Hox-A, -B*, and *-C* clusters (*Table 1*), which were validated by reverse-transcription polymerase chain reaction (RT-PCR) (*Figure 2G*). These results suggest that there is a good correlation between the intranuclear distribution of Nup98-HoxA9 and its function in transcription, since *Hox* gene cluster regions are known to be organized as facultative heterochromatin (*Trojer and Reinberg, 2007*).

## Influence of Nup98 FG repeats on the activation of *Hox* genes cannot be mimicked by FG repeat domains of other Nup proteins

To find out how *Hox* genes could be selectively induced by the Nup98-HoxA9 fusion in ES cells, we first focused on Nup98FG repeats, since all Nup98-fusion proteins, including Nup98-HoxA9, contain dense FG repeats of Nup98 at their N-termini. On the other hand, it is known that other nucleoporins containing such dense FG repeat domains are also implicated in oncogenic transformation (*Kasper et al., 1999*, *Xu and Powers, 2009*). Therefore, we created chimeric Nup-fusions by replacing Nup98 FG repeats with those from hNup153FG (A.A.1118–1475) and hNup214FG (A.A.1605–2090), which have been demonstrated to be functionally exchangeable in experiments that addressed the transforming ability in NIH3T3 cells (*Kasper et al., 1999*). When these chimeric proteins (Nup153-HoxA9 and Nup214-HoxA9) were expressed in HeLa cells, they were not distributed as distinct fine nuclear dots observed in the case of the Nup98-HoxA9 fusion (*Figure 3—figure supplement 1*), in agreement with a previous report (*Xu and Powers, 2013*).

We next isolated stable ES cell clones expressing these fusions. As shown in *Figure 4—figure supplement 2*, we found that, in contrast to the case with Nup98-HoxA9, other Nup-fusions were distributed more evenly in the nucleus (although note that a few dots within the nucleus were seen with the Nup214-HoxA9 fusion). Next, the effect of these Nup-fusions on the expression of *Hox* genes was monitored in established stable ES cell lines. As shown in *Figure 3*, the expression of Nup153-HoxA9 or Nup214-HoxA9 did not affect the expression of *Hox* cluster genes. Of note, we constantly observed substantial numbers of alkaline phosphatase-positive colonies of Nup153-HoxA9-expressing ES cells in the absence of LIF (*Figure 2D* and *Figure 2—figure supplement 3*), which may be related to the function of Nup153 in maintaining stem cell pluripotency in ES cells, as demonstrated in a recent study (*Jacinto et al., 2015*). We also established stable clones expressing full-length HoxA9. Gene expression analysis revealed that HoxA9 expression did not induce the activation of *Hox* cluster genes (*Figure 3*). These results underscore the physiological significance of Nup98 FG repeats in the dysregulation of *Hox* cluster gene expression.

## Nup98-HoxA9 dots are maintained through the association with Crm1

What makes Nup98FG repeats differ from Nup153FG or Nup214FG? We speculated that cohesive properties of Nup98FG (*Patel et al., 2007*; *Xu and Powers, 2013*; *Yamada et al., 2010*; *Schmidt and Gorlich, 2015*) required to form nuclear aggregates (*Griffis et al., 2002*; *Oka et al., 2010*; *Patel et al., 2007*) containing Crm1 (*Takeda et al., 2010*, *Oka et al., 2010*) may be important

**Table 1.** List of the top 32 genes that were upregulated both in Nup98-HoxA9 clone #1 and Nup98-HoxA9 clone#9.

| Description | Fold change (log2) | | |
| --- | --- | --- | --- |
| | 98H#1 /EB3 | 98H#9 /EB3 | HoxA9-Ct#22/EB3 |
| Mus musculus homeobox A3 (Hoxa3), mRNA [NM_010452] | 9.15 | 9.37 | 0.05 |
| Mus musculus homeobox A10 (Hoxa10), transcript variant 1, mRNA [NM_008263] | 8.17 | 7.61 | 0.05 |
| Mus musculus 2 days pregnant adult female oviduct cDNA, RIKEN full-length enriched library, clone:E230011F24 [AK053996] | 7.60 | 7.71 | 0.08 |
| Mus musculus homeobox A7 (Hoxa7), mRNA [NM_010455] | 7.69 | 7.44 | 0.32 |
| Mus musculus homeobox A11 (Hoxa11), mRNA [NM_010450] | 7.87 | 6.95 | 0.06 |
| Mus musculus homeobox C6 (Hoxc6), mRNA [NM_010465] | 8.46 | 5.99 | 0.05 |
| Mus musculus homeobox A5 (Hoxa5), mRNA [NM_010453] | 7.08 | 7.11 | 0.06 |
| Mus musculus blastocyst blastocyst cDNA, RIKEN full-length enriched library, clone:I1C0031F10 product [AK145700] | 6.16 | 7.10 | 1.26 |
| Mus musculus homeobox A9 (Hoxa9), mRNA [NM_010456] | 6.60 | 6.64 | 0.09 |
| predicted gene 3395 [Source:MGI Symbol;Acc:MGI:3781573] [ENSMUST00000172100] | 5.95 | 6.77 | 1.45 |
| Mus musculus blastocyst blastocyst cDNA, RIKEN full-length enriched library, clone:I1C0015F22 product [AK145555] | 5.87 | 6.76 | 0.51 |
| Mus musculus HOXA11 antisense RNA (non-protein coding) (Hoxa11as), non-coding RNA [NR_015348] | 6.59 | 5.78 | -0.34 |
| Mus musculus blastocyst blastocyst cDNA, RIKEN full-length enriched library, clone:I1C0027E24 product [AK167004] | 5.54 | 6.46 | 0.75 |
| Mus musculus homeobox B8 (Hoxb8), mRNA [NM_010461] | 6.56 | 5.38 | 0.06 |
| Mus musculus blastocyst blastocyst cDNA, RIKEN full-length enriched library, clone:I1C0015H22 product [AK166824] | 5.53 | 6.36 | 1.07 |
| Mus musculus RCB-0559 K-1. F1 cDNA, RIKEN full-length enriched library, clone: G430049J08 product [AK144159] | 5.19 | 6.19 | 0.37 |
| Mus musculus homeobox A4 (Hoxa4), mRNA [NM_008265] | 5.24 | 5.49 | 0.29 |
| Mus musculus blastocyst blastocyst cDNA, RIKEN full-length enriched library, clone:I1C0037K09 product [AK145750] | 4.93 | 5.72 | 0.83 |
| Mus musculus homeobox C4 (Hoxc4), mRNA [NM_013553] | 6.19 | 4.29 | 1.28 |
| Mus musculus homeobox B4 (Hoxb4), mRNA [NM_010459] | 5.27 | 4.61 | -0.33 |
| Mus musculus homeobox A6 (Hoxa6), mRNA [NM_010454] | 4.95 | 4.82 | -1.14 |
| Mus musculus homeobox B7 (Hoxb7), mRNA [NM_010460] | 5.40 | 4.16 | -0.75 |
| Mus musculus homeobox B2 (Hoxb2), mRNA [NM_134032] | 4.77 | 4.30 | 0.44 |
| Mus musculus cystatin 13 (Cst13), mRNA [NM_027024] | 3.72 | 5.31 | 2.99 |
| Mus musculus brain and acute leukemia, cytoplasmic (Baalc), mRNA [NM_080640] | 3.57 | 5.44 | 1.17 |
| Mus musculus homeobox A2 (Hoxa2), mRNA [NM_010451] | 4.47 | 4.37 | 0.16 |
| Mus musculus CAP, adenylate cyclase-associated protein, 2 (yeast) (Cap2), mRNA [NM_026056] | 4.48 | 4.30 | 1.16 |
| RIKEN cDNA 5730446D14 gene [Source:MGI Symbol;Acc:MGI:1913890] [ENSMUST00000155922] | 4.30 | 4.44 | 0.06 |
| Mus musculus olfactory receptor 161 (Olfr161), mRNA [NM_146860] | 3.50 | 5.06 | 0.56 |
| Mus musculus carbonyl reductase 2 (Cbr2), mRNA [NM_007621] | 4.25 | 3.97 | 0.06 |
| Mus musculus homeobox A1 (Hoxa1), mRNA [NM_010449] | 3.86 | 4.13 | -0.11 |
| Mus musculus Scm-like with four mbt domains 2 (Sfmbt2), transcript variant 3, mRNA [NM_001198809] | 3.58 | 3.65 | 2.69 |

cDNA, complementary DNA; mRNA, messenger RNA.

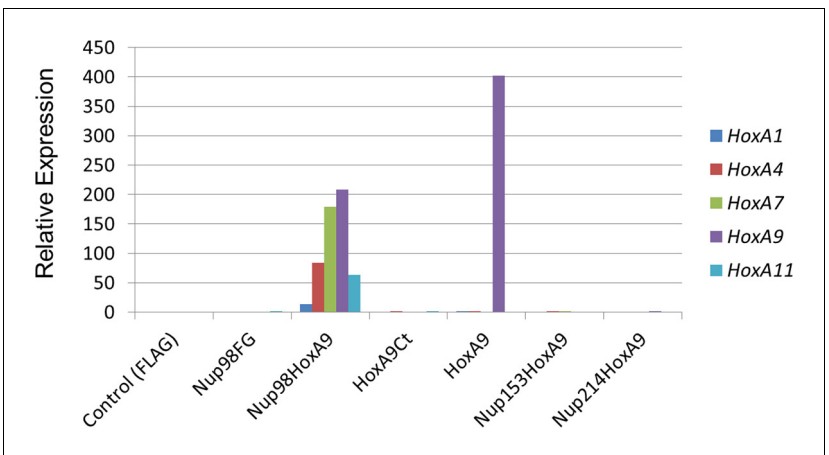

**Figure 3.** Functional characterization of various NupFG-HoxA9 fusions. qPCR analysis of *Hox-A* cluster gene expression in various ES cell lines expressing the NupFG-HoxA9 fusion, Nup98FG, or HoxA9. *GAPDH* was used as a reference gene. ES, embryonic stem ; qPCR, quantitative polymerase chain reaction.
The following figure supplement is available for figure 3:

**Figure supplement 1.** Subcellular localization of various NupFG-HoxA9 fusions.

for the pathogenic function. Indeed, the ectopic expression of Nup98-HoxA9 in HeLa cells caused sequestration of nucleoplasmic and nuclear envelope-localized Crm1 on Nup98-HoxA9 dots within the nucleus (*Figure 4A*, top). When we stained Nup98-HoxA9 ES cells, we also noticed partial recruitment of Crm1 to FLAG-Nup98-HoxA9 dots (*Figure 4A*, bottom). We have previously demonstrated that treatment with leptomycin B (LMB), a specific inhibitor of the cargo binding of Crm1 (*Kudo et al., 1998*), causes a dynamic redistribution of Nup98 nuclear dots (*Oka et al., 2010*). This finding indicated that Nup98-Crm1 association is dependent on the cargo binding status of Crm1; therefore, we examined whether such properties of Nup98FG were still preserved in Nup98-HoxA9. When Nup98-HoxA9 ES cells were treated with LMB for 2 hr, the dots disappeared (*Figure 4B*). The disappearance of dots was concomitant with a modest decrease in the protein level (*Figure 4—figure supplement 1*). In contrast, LMB treatment did not affect the localization of full-length HoxA9, Nup153-HoxA9, and the majority of Nup214-HoxA9, except for a few nuclear dots of Nup214-HoxA9 that disappeared upon LMB treatment (*Figure 4—figure supplement 2*). These results indicate that Crm1 plays a critical role in the maintenance of Nup98-HoxA9 dots in ES cells.

## Crm1 plays an important role in Nup98-HoxA9 functions

Next, to examine whether the association between Nup98-HoxA9 and Crm1 is required for the Nup98-HoxA9-mediated *Hox* gene activation, we treated Nup98-HoxA9 ES cells with LMB and monitored its effects on gene expression. As shown in *Figure 4C*, LMB treatment caused a significant downregulation of *Hox* cluster genes, while the expression of other genes, such as *Oct4* and *Nanog*, was not much affected. Therefore, these results indicate that the interaction of Nup98-HoxA9 with Crm1 is necessary for the selective *Hox* gene activation.

## Nup98-HoxA9 selectively binds to four *Hox* cluster regions

To further investigate how Nup98-HoxA9 is involved in the *Hox* cluster gene regulation, we performed chromatin immunoprecipitation sequencing (ChIP-seq) analysis using Nup98-HoxA9 ES cells. Interestingly, we found a strong accumulation of FLAG-Nup98-HoxA9 on chromosomal regions of four *Hox* clusters, *Hox-A, B, C,* and *D*. Especially, three of these, *Hox-A, Hox-B* and *Hox-C* cluster regions showed the highest peaks in the whole genome (*Figure 5A*). In particular, the *Hox-A* cluster region showed a single dominant peak in chromosome 6 (*Figure 5C*). Examination of individual *Hox* cluster regions further revealed a selective accumulation of FLAG-Nup98-HoxA9 in the four *Hox* cluster regions (*Figure 5D*). It is noteworthy that, as most of the genes in the *Hox-D* cluster were not

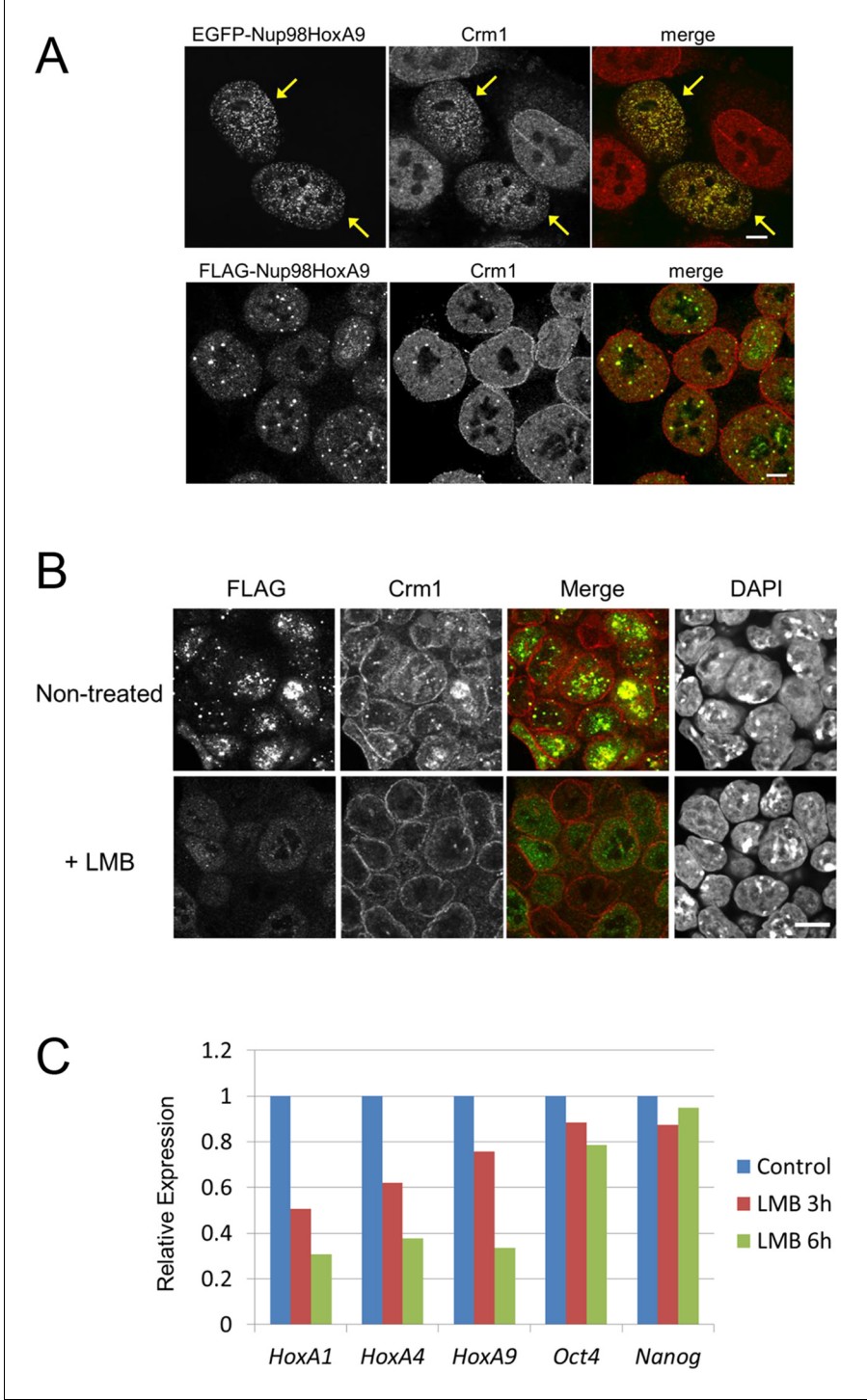

**Figure 4.** Association between Nup98-HoxA9 and Crm1 is critical for the *Hox* Gene activation mediated by Nup98-HoxA9. (**A**) *Top panel*: Nup98-HoxA9 interacts and sequesters Crm1 onto Nup98-HoxA9 dots. HeLa cells were transfected with the EGFP-Nup98-HoxA9 expressing plasmid. After 24 hr, cells were fixed and stained with an anti-Crm1 antibody. Arrows indicate the cells transfected. *Bottom panel*: Nup98-HoxA9 ES cells were fixed and co-stained with anti-FLAG (M2) and anti-Crm1 antibodies. Merged image of FLAG (green) and Crm1 (red) is shown. Bar, 5 μm. (**B**) The effect of LMB treatment on the cellular localization of Nup98-HoxA9. Nup98-HoxA9 ES cells were cultured either in the presence or absence of 5 nM LMB for 2 hr, fixed and stained with antibodies against FLAG (M2) and Crm1. Merged images of FLAG (green) and Crm1 (red) are shown. Nuclei were stained with DAPI. Bar, 10 μm. (**C**) Effect of LMB treatment on the regulation of *Hox* cluster genes. Nup98-HoxA9 ES cells were cultured in the presence or absence of 5 nM LMB for 3 or 6 hr and the expression of indicated genes was analyzed by qPCR. *GAPDH* was used as a reference gene. EGFP, enhanced green fluorescent protein; LMB, leptomycin B; qPCR, quantitative polymerase chain reaction.

*Figure 4. continued on next page*

*Figure 4. Continued*

The following figure supplements are available for figure 4:

**Figure supplement 1.** Effect of LMB treatment on the FLAG-Nup98-HoxA9 protein level.

**Figure supplement 2.** Effect of LMB treatment on the cellular localization of various NupFG-HoxA9 fusions.

upregulated by Nup98-HoxA9, the significance of its binding to this region remains unknown. ChIP-qPCR (quantitative polymerase chain reaction) analysis (*Figure 5E*) confirmed that a significantly increased amount of FLAG-Nup98-HoxA9 was bound to the *Hox-A* cluster, but not to the restricted regions, as represented by the intergenic region (between *HoxA2* and *HoxA3*), promoter (*HoxA4*), or 5'UTR (*HoxA9*) regions of *Hox-A* cluster genes.

## Chromosomally prebound Crm1 recruits Nup98-HoxA9 to *Hox* cluster regions

Next, we examined whether Crm1 could bind *Hox* cluster regions using ChIP-seq analysis. Unexpectedly, we found that in parental ES cells, the endogenous Crm1 bound to all *Hox* regions including *Hox-A, B, C,* and *D* (*Figure 5B–D*), which were almost the same sites as those occupied by Nup98-HoxA9. Moreover, the binding of Crm1 to *Hox* cluster regions was significantly elevated in Nup98-HoxA9 ES cells (*Figure 5B–E*). Furthermore, LMB treatment induced a drastic decrease of Nup98-HoxA9 binding to *Hox* clusters (*Figure 5B–E*). These results indicate that Nup98-HoxA9 forms higher order nuclear structures containing Crm1 on *Hox* cluster regions to induce *Hox* gene activation.

## Genome-wide characterization of Nup98-HoxA9 and Crm1 binding sites

Next, we characterized the chromatin binding sites of FLAG-Nup98-HoxA9 and their overlap with Crm1-binding sites on a genome-wide scale. Aggregation plot revealed that the peak of FLAG-Nup98-HoxA9 binding site was severely diminished but not shifted by LMB treatment (*Figure 6A*, left top). Furthermore, FLAG-Nup98-HoxA9 was preferentially targeted to the chromatin sites that originally bound Crm1 (the Crm1 binding site in control ES cells) (*Figure 6A*, right top). Aggregation plots of the Crm1 binding signal revealed that the peak of Crm1 binding was not shifted, but rather significantly enhanced by the expression of FLAG-Nup98-HoxA9 (*Figure 6A*, left bottom). In addition, the Crm1 signal also accumulated at the FLAG-Nup98-HoxA9 binding site (*Figure 6A*, right bottom). Statistical analyses confirmed these results (*Figure 6B*). We also performed an analysis of the target genes that bound FLAG-Nup98-HoxA9 or Crm1 (*Figure 6C*, *Figure 6—figure supplement 1*). As expected, the number of FLAG-Nup98-HoxA9 target genes was severely decreased by LMB treatment (*Figure 6—figure supplement 1*, top left). On the other hand, the number of Crm1 target genes were significantly increased by the expression of FLAG-Nup98-HoxA9; however, the majority of these genes were not overlapping (*Figure 6C*, *Figure 6—figure supplement 1*, top right), examples of which were observed in the vicinity of *Hox*-cluster regions (*Figure 5D*, see green arrows near *Hox-A (Skap2)* or *Hox-C* cluster). In these regions, it is likely that FLAG-Nup98-HoxA9 first binds to chromatin loci where Crm1 is not present, and then, Crm1 is recruited to FLAG-Nup98-HoxA9-bound sites, which results in the increase of Crm1-target genes on a whole genome level (*Figure 6C*, *Figure 6—figure supplement 1*, compare bottom two panels). Therefore, these results demonstrate that both chromatin-bound Crm1 and FLAG-Nup98-HoxA9 can recruit each other.

## Binding of Nup98-HoxA9 and Crm1 to chromatin is sensitive to TSA treatment

Our results so far suggest that Nup98-HoxA9 is targeted to Crm1 that associates with specific chromatin modifications. Therefore, we examined the effects of various epigenetic inhibitors on nuclear Nup98-HoxA9 dots. While most of these agents only showed a minor effect on the localization pattern of Nup98-HoxA9 (*Figure 7A*), trichostatin A (TSA) treatment caused a drastic redistribution of FLAG-Nup98-HoxA9, a significant increase in the number and a significant decrease in the size of dots, which mimicked that observed in HeLa cells (*Figure 7B*), concomitant with the redistribution of endogenous Crm1 (*Figure 7—figure supplement 1*). We also confirmed that TSA treatment caused

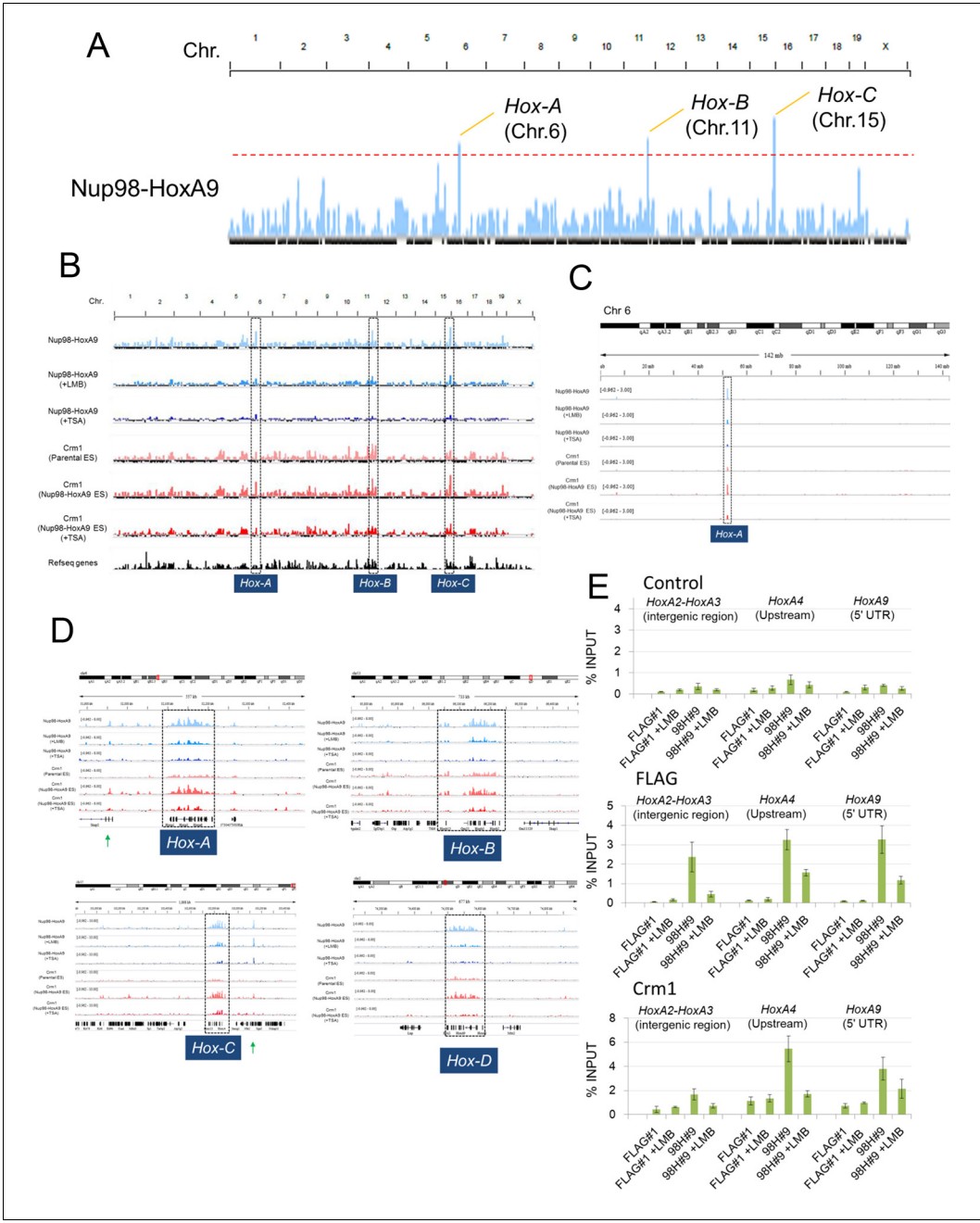

**Figure 5.** Nup98-HoxA9 is highly selectively targeted to *Hox* cluster regions through the chromatin-associated Crm1. (A–D) ChIP-seq analysis of Nup98-HoxA9 or Crm1 (A, B: whole genome; C: chromosome 6; D: *Hox* cluster regions). The parental ES or Nup98-HoxA9 ES cells were cultured either in the absence or presence of LMB (5 nM, 2 hr) or TSA (50 nM, 24 hr) and used for ChIP-seq analysis. Also in (D) are regions that show Crm1 signals only when Nup98-HoxA9 is expressed (eg. green arrows point to regions next to *Hox-A* and *Hox-C*). (E) ChIP-qPCR analysis of FLAG-Nup98-HoxA9 and Crm1. Data are mean values ± standard error of the mean of three independent experiments. ChIP-seq, ChIP sequencing; ES, embryonic stem; LMB, leptomycin B; qPCR, quantitative polymerase chain reaction; TSA, trichostatin A.

The following figure supplement is available for figure 5:

**Figure supplement 1.** Binding of Nup98FG or Nup98HoxA9 to *Hox-A* cluster region.

an upregulation of the FLAG-Nup98-HoxA9 protein and mRNA (***Figure 7—figure supplement 2***),

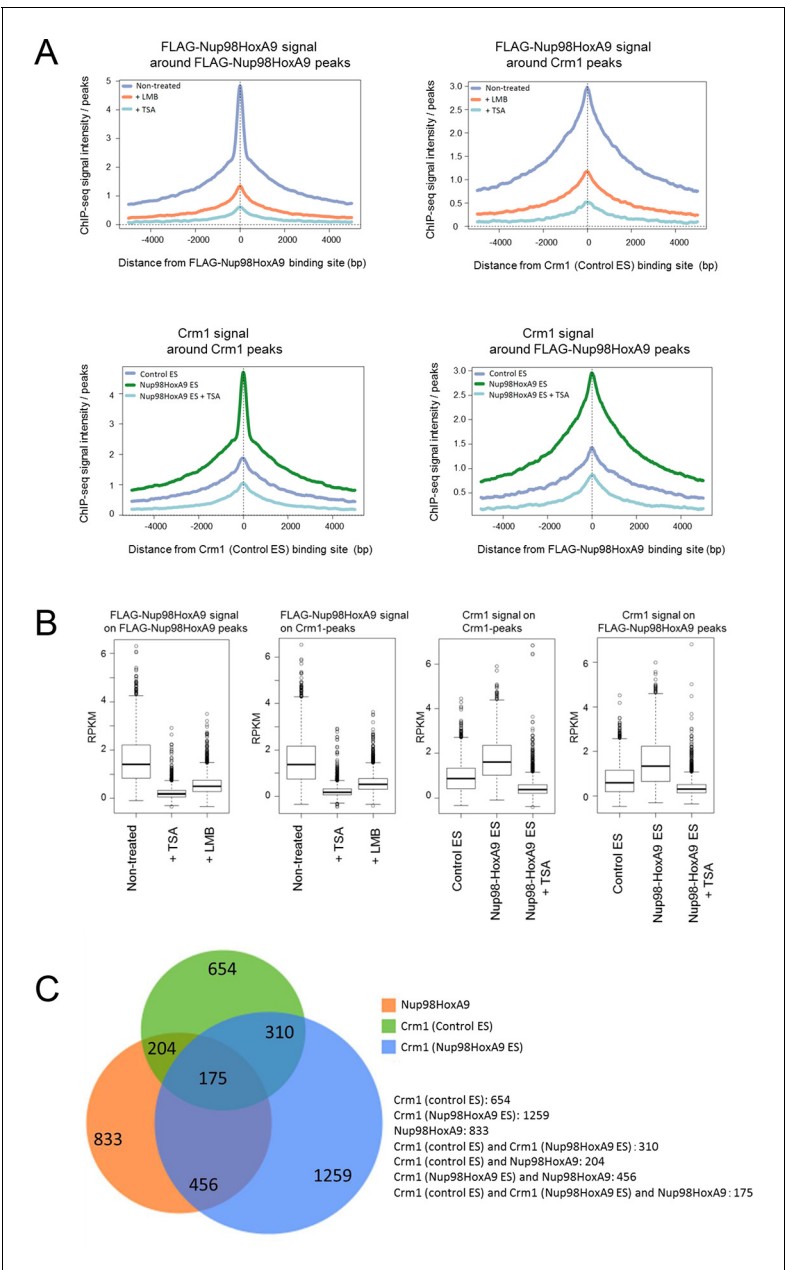

**Figure 6.** Genome-wide analysis of FLAG-Nup98-HoxA9 and Crm1 binding sites. (**A**) Aggregation plots of FLAG-Nup98-HoxA9 and Crm1 binding sites. FLAG-Nup98-HoxA9 binding signals in untreated, LMB-treated, and TSA-treated cells are mapped against either FLAG-Nup98-HoxA9 (top, left) or Crm1 binding sites (top, right) of untreated cells. Crm1 binding signals in control, Nup98-HoxA9 ES cells, and TSA-treated Nup98-HoxA9 ES cells were mapped against either Crm1 (bottom, left) or FLAG-Nup98-HoxA9 binding sites (bottom, right) of untreated cells. (**B**) Box plot showing FLAG-Nup98-HoxA9 signals (RPKM, reads per kilobase per million mapped reads) in untreated, TSA-treated, and LMB-treated cells either around FLAG-Nup98-HoxA9 binding sites (upstream 2 kb to downstream 2 kb) in untreated Nup98-HoxA9 ES cells or around Crm1 binding sites (upstream 2 kb to downstream 2 kb) in control ES cells, or Crm1 signals (RPKM) in control ES, Nup98-HoxA9 ES, and TSA-treated Nup98-HoxA9 ES cells around Crm1 binding sites (upstream 2 kb to downstream 2kb) in control ES cells, or around FLAG-Nup98-HoxA9 binding sites in untreated Nup98-HoxA9 ES cells. (**C**) Venn diagrams showing the overlap between genes associated with FLAG-Nup98-HoxA, Crm1 in control ES, and Crm1 in Nup98-HoxA9 ES. Genes that contained FLAG-Nup98-HoxA or Crm1 binding site(s) within 5000 bp upstream and 1000 bp downstream of the TSS were analyzed. The numbers indicate the number of genes in each category. ES, embryonic stem; LMB, leptomycin B; TSA, trichostatin A; TSS, transcription start site.

The following figure supplement is available for figure 6:

**Figure supplement 1.** Venn diagrams showing the overlap between genes associated with FLAG-Nup98-HoxA9 and/or Crm1 in various cell lines or at different culture conditions.

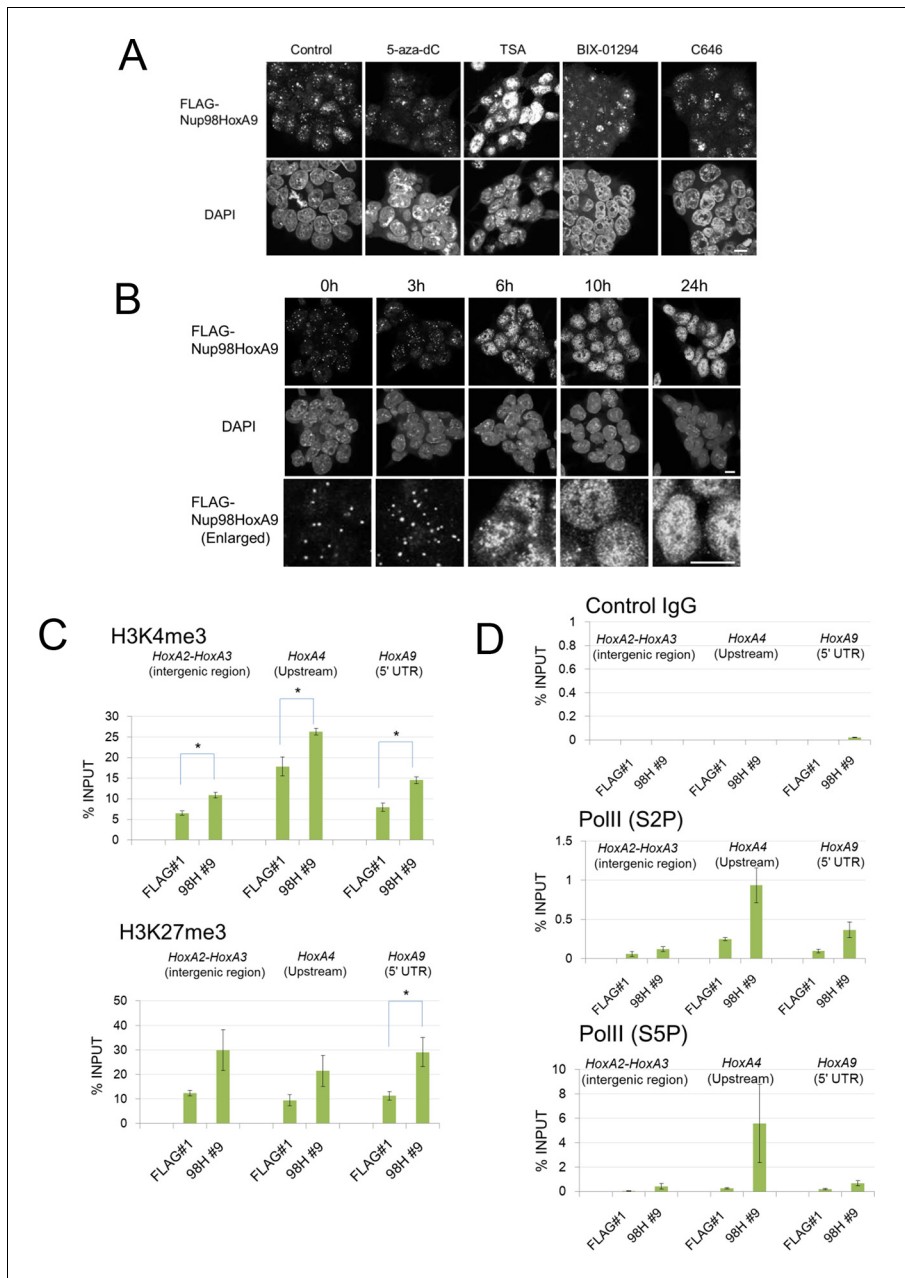

**Figure 7.** Epigenetic status and chromatin binding of Nup98-HoxA9/Crm1. (**A**) Effect of various epigenetic inhibitors on the localization of Nup98-HoxA9. Epigenetic effectors; 5-aza-dC (an inhibitor of DNA methyltransferase), TSA (an inhibitor of histone deacetylases), BIX-1294 (an inhibitor of the G9a histone methyltransferase), and C646 (an inhibitor of the histone acetyltransferase p300). Immunostaining of FLAG-Nup98-HoxA9 ES cells was performed upon treatment with either 5-aza-dC (2 μM), TSA (100 nM), BIX-01294 (10 μM) or C646 (25 μM) for 24 hr. Nuclei were stained with DAPI. Bar, 10 μm. (**B**) Time-course of the Nup98-HoxA9 localization upon TSA (50 nM) treatment. Nuclei were stained with DAPI. Bar, 10 μm. (**C**) The effect of Nup98-HoxA9 expression on the histone modification of the *Hox-A* cluster region. ChIP-qPCR analysis was performed using an anti-H3K4me3 or an anti-H3K27me3 antibody. Data are mean values ± standard error of the mean of three independent experiments. Statistical significance was evaluated with the Student t test. *p < 0.05. (**D**) RNAPII recruitment at *Hox* cluster region. ChIP-qPCR analysis was performed using anti-PolII (S2P) or anti-PolII (S5P) antibody. Data are mean values ± standard error of the mean of three independent experiments. 5-aza-dC, 5-Aza-2'-deoxycytidine; ChIP, chromatin immunoprecipitation; DAPI, 4',6-diamidino-2-phenylindole; qPCR, quantitative polymerase chain reaction; TSA, trichostatin A.

The following figure supplements are available for figure 7:

**Figure supplement 1.** Effect of TSA treatment on the subcellular localization of Crm1.

**Figure supplement 2.** Effect of TSA treatment on the expression of FLAG-Nup98-HoxA9.

*Figure 7. Continued*

**Figure supplement 3.** Effect of TSA treatment on Crm1 binding to the *Hox-A* cluster region in control ES cells.

**Figure supplement 4.** Effect of TSA treatment on histone modifications of the *Hox-A* cluster region.

which occurred presumably due to the reactivation of partially silenced transgenes cloned in the Tol2 transposon vector. A similar phenomenon was also observed in the case of transgenes created by the Sleeping Beauty transposon (*Garrison et al., 2007*). Next, we examined the effects of TSA treatment on the genome-wide chromatin binding of Crm1 or FLAG-Nup98-HoxA9. Unexpectedly, despite the increase in the FLAG-Nup98-HoxA9 protein level, TSA treatment caused a drastic decrease in the binding of Nup98-HoxA9 and Crm1 to chromatin (*Figure 6A,B*), which was also obvious in *Hox* cluster regions (*Figure 5B–D*). Furthermore, our ChIP-qPCR analysis showed that the binding of endogenous Crm1 to *Hox-A* cluster in control ES cells was also sensitive to TSA treatment (*Figure 7—figure supplement 3*).

## Effect of TSA treatment on the chromatin structure of *Hox* cluster regions

It is known that TSA treatment causes differential effects on the epigenetic status of chromatin depending on the dosage, duration, cell-type, and target genes. Interestingly, it is known that treatment of ES cells with TSA at a concentration of 50 nM drastically affects bivalent histone modifications H3K4me3 (upregulation) and H3K27me3 (downregulation), in addition to the acetylation (*Karantzali et al., 2008*). Since bivalent histone marks are critical for the regulation of *Hox* cluster genes in ES cells (*Mikkelsen et al., 2007*, *Pan et al., 2007*, *Zhao et al., 2007*), we speculate that TSA treatment may dramatically modulate the epigenetic status of *Hox* cluster regions. Indeed, as shown in *Figure 7—figure supplement 4*, TSA treatment caused a moderate upregulation of H3K4me3 and a robust decrease of H3K27me3 on the promoter region of *Hox-A* cluster genes, which is in line with a previous report (*Karantzali et al., 2008*). These results suggest that inactive histone marks, including H3K27me3, are important for the recruitment of Crm1 and FLAG-Nup98-HoxA9 to the *Hox-A* cluster region.

## Nup98-HoxA9 binding causes epigenetic modifications and recruitment of RNA polymerase II onto *Hox-A* cluster

We then compared the epigenetic status of the *Hox-A* region in Nup98-HoxA9 and parental ES cells. We revealed that Nup98-HoxA9 ES cells exhibited significantly higher levels of H3K4me3, an active histone modification, in all tested *Hox* regions (*Figure 7C*). On the other hand, enhanced levels of H3K27me3 were significant only on the *HoxA9* 5'UTR exon (*Figure 7C*). Furthermore, ChIP-qPCR analysis of phospho-CTD isoforms of RNA polymerase II (RNAPII) revealed that the binding of both S5P (initiating) and S2P (elongating) forms of RNAPII to the *Hox-A* cluster regions increased in the Nup98-HoxA9 ES cells compared with parental ES cells (*Figure 7D*).

Collectively, our results suggest that Crm1 may associate with *Hox* genes via inactive histone marks, such as H3K27me3, and subsequently, Nup98-HoxA9 may form a local higher order chromatin structure on Crm1 binding sites to further recruit active histone modifier(s) and RNAPII to induce a selective upregulation of *Hox* genes.

## Discussion

Our findings reveal that Nup98-HoxA9 is selectively recruited to *Hox* cluster regions via its interaction with Crm1 to induce gene expression in ES cells. Although activation of several *Hox* genes by Nup98-HoxA9 has been previously reported in other types of cells (*Calvo et al., 2002*; *Takeda et al., 2006*; *Ghannam et al., 2004*), those effects were relatively weak and not highly specific, which is in contrast to the pronounced changes observed in our present study. We believe that such variable outcomes could stem from two possible reasons. First, the chromatin structure, epigenetic status or chromatin-bound protein complexes of *Hox* cluster regions may be dissimilar in

different cell types, causing varying susceptibility to Nup98-HoxA9-mediated chromatin modifications. Second, the association status of Crm1 with *Hox* cluster gene loci, which is a prerequisite for the selective deregulation of *Hox* cluster genes, may be differentially modulated in different cell types, causing cell type-specific variability in the Nup98-HoxA9-mediated gene expression.

Notably, although both FG repeats of Nup153 and Nup214 are known to interact with Crm1 (*Roloff et al., 2013*; *Nakielny et al., 1999*; *Fornerod et al., 1997*), we found that these fusions, Nup153-HoxA9 and Nup214-HoxA9, failed to activate *Hox* cluster genes. Thus, our results suggest that the ability of Nup98 FG repeats to form nuclear dot-like structures is critical in *Hox* gene activation. We propose a model whereby the interaction of Nup98-HoxA9 with prebound Crm1 induces the alteration of chromatin structures specifically on *Hox* cluster regions and this modification, in turn, causes robust activation of gene expression.

It is known that *Hox* clusters in ES cells harbor a distinctive long-range histone modification signature called the bivalent motif, which possesses both transcriptionally active H3K4me3 and repressive H3K27me3 marks (*Bernstein et al., 2006*; *Mikkelsen et al., 2007*). This modification is maintained through the activity of polycomb complexes (*Schwartz and Pirrotta, 2007*). Our results demonstrate that Nup98-HoxA9/Crm1 binding to *Hox* region is sensitive to TSA treatment, which results in a robust decrease of the H3K27me3 mark on *Hox* cluster genes. Moreover, the expression of Nup98-HoxA9 causes a significant increase of the active histone mark H3K4me3 on *Hox* cluster region genes. Previously, it was reported that the FG repeat region of Nup98 interacts with histone acetyltransferases CBP and p300 (*Kasper et al., 1999*). In addition, MBD-R2/NSL and Trx histone modifying complexes have been demonstrated to interact with full-length Nup98 in fly (*Pascual-Garcia et al., 2014*). Thus, we speculate that Nup98-HoxA9 first binds to facultative heterochromatin and then recruits histone and/or chromatin modifying enzymes to induce active histone modifications and a subsequent activation of *Hox* cluster genes. Importantly, in CD133+ hematopoietic stem cells, histone modifications of *Hox-A* and *Hox-B* loci are also bivalent (*Cui et al., 2009*), suggesting that *Hox* cluster genes are similarly regulated both in ES and hematopoietic progenitor cells.

Selective activation of *Hox* genes has also been reported in other Nup98-fusions, Nup98-NSD1 and Nup98-Jarid1a, in myeloid progenitor cells (*Wang et al., 2007*, *Wang et al., 2009*). Mechanistically, Nup98-NSD1 binds to the promoter region of *Hox-A* cluster genes and modifies histones via its H3K36 methyltransferase activity (*Wang et al., 2007*). Similarly, Nup98-Jarid1a binds to di- or tri-methylated H3K4 of the *Hox-A* cluster region through its plant homeodomain (PHD) finger domain and causes deregulation of *Hox* genes activation (*Wang et al., 2009*). However, it remained unclear how the expression of these Nup98-fusions actually causes selective activation of *Hox* genes. Our results suggest that chromosome prebound Crm1 may recruit Nup98-NSD1, Nup98-Jarid1a, and possibly other Nup98-fusions to specific *Hox* cluster genes loci through its interaction with the Nup98FG repeat, which commonly exists in all Nup98-fusions. Unexpectedly, our data showed that Nup98FG could only weakly bind to *Hox-A* cluster region (*Figure 5—figure supplement 1*). However, since Nup98FG dots are frequently localized within the nucleolus (data not shown), and Nup98FG is more prone to form nuclear aggregates than Nup98-HoxA9 with only a weak nucleoplasmic diffuse staining (see *Figure 2B*), we speculate that Nup98FG is hardly accessible to the genomic DNA, including the *Hox* cluster region, when expressed by itself.

Notably, a recent report (*Conway et al., 2014*) also demonstrated that Crm1 binds to the *HoxA9* and *HoxA10* gene regions, both in human leukemia cell line and immortalized mouse embryonic fibroblast cell line. *Conway et al. (2014)* further showed that Crm1 recruits CALM-AF10 fusion through its interaction with the nuclear export signal on CALM to activate *Hox* gene expression. Thus, the association of Crm1 with *Hox* loci could be a common molecular basis for aberrant *Hox* gene dysregulation mediated by numerous leukemic fusions.

## Materials and methods

### Cell culture

HeLa cells and NIH3T3 cells were cultured in the Dulbecco's modified Eagle's medium (DMEM; Sigma-Aldrich, St. Louis, MO) supplemented with 10% fetal bovine serum (FBS). The parental EB3 ES cells (*Niwa et al., 2002*; *Ogawa et al., 2004*) and their derivatives were cultured on gelatin-coated dishes in DMEM supplemented with 10% FBS, 10 mM MEM non-essential amino acids (Life

Technologies, Carlsbad, CA), 100 mM sodium pyruvate (Life Technologies), 0.1 mM β-mercaptoethanol (Life Technologies), and LIF at 37°C in 5% $CO_2$ atmosphere.

## Antibodies

The following mouse monoclonal antibodies against specific histone modifications were used in this study: anti-H3K4me3, anti-H3K27me3, anti-H3K9me2, and anti-H3K9me3 (*Kimura et al., 2008*). Rat monoclonal Pol II antibodies to site specific phosphorylation used in this study were as follows; Anti-Pol II (S2P), Anti-Pol II (S5P) (*Odawara et al., 2011*). The anti-FLAG (M2, #F1804) and anti-FLAG (polyclonal, #F7425) antibodies were purchased from Sigma-Aldrich. The anti-Crm1antibody (#NB100-79802) was purchased from Novus Biologicals (Littleton, CO). The anti-Nup98 antibody is as described (*Fukuhara et al., 2005*).

## Plasmids

To generate EGFP-Nup98FG, EGFP-Nup98-HoxA9, and EGFP-HoxA9-Ct mammalian expression vectors, the coding sequences for N-terminus of human Nup98 (A.A. 1–469), C-terminus of HoxA9 (A.A. 164–272), Nup98-HoxA9 (combination of N-terminus of human Nup98 and C-terminus of HoxA9 were amplified and cloned into the pEGFP-C1 vector (Clontech, Mountain View, CA). Sequences were verified by sequencing analysis. The Tol2-based Nup-fusion expression vectors were generated by inserting coding sequences for Nup98-HoxA9, Nup98FG (A.A. 1–469), HoxA9-Ct (A.A. 164–272), HoxA9 (full length), Nup214-HoxA9 (a fusion between Nup214 (A.A.1605–2090) and HoxA9-Ct), Nup153-HoxA9 (a fusion between Nup153 (A.A.1118–1475) and HoxA9-Ct) into the pT2A-CMH, a Tol2 transposon-based vector containing a multiple cloning site (*Oka et al., 2013*). The coding regions of the inserted cDNA molecules were verified by sequencing analysis.

## Transfection and generation of stable clones

For transient transfection assays, cells were plated into 35-mm dishes and transfected with plasmid DNA using lipofectamine 2000 (Life Technologies). Generation of stable ES cell lines expressing a transgene was performed as described previously (*Oka et al., 2013*). Briefly, ES cells were co-transfected with pCAGGS-m2TP, which is an expression vector for Tol2 transposase, and the Tol2 transposon-based pT2A-CMH expression vector containing various inserted cDNA sequences. After 2 d, cells were re-plated into the ES-LIF medium containing hygromycin B (200 µg/ml). The medium was changed every other day until the appearance of colonies. Colonies were picked and the expression product of the transgenes was confirmed by western blotting and immunofluorescence staining.

## Immunofluorescence staining and confocal microscopy

Cells were grown on coverslips and fixed with 3.7% formaldehyde in phosphate-buffered saline (PBS) for 15 min at room temperature. After treatment with 0.5% Triton X-100 in PBS for 5 min, the cells were incubated in blocking buffer (3% skim milk in PBS) for 30 min and again incubated with primary antibodies overnight at 4°C. After washing with PBS, cells were incubated with Alexa Flour 488– and/or 594– conjugated secondary antibodies (Life Technologies) for 30 min. The cells were then stained with DAPI and the coverslips were mounted with Vectashield (Vector Laboratories, Burlingame, CA). Images were acquired using FV1000 (Olympus) or SP8 (Leica) confocal microscopes.

## Microarray analysis

Microarray analysis was performed using total RNA prepared from indicated ES cell clones using the TRIzol reagent (Invitrogen Carlsbad, CA) as follows: cyanine-3 (Cy3) labeled complementary RNA (cRNA) was prepared from 0.1 µg total RNA using the Low Input Quick Amp Labeling Kit (Agilent Technologies, Santa Clara, CA) according to the manufacturer's instructions, followed by RNAeasy column purification (Qiagen, Valencia, CA). Cy3-labeled cRNA (0.6 µg) was fragmented and hybridized to SurePrint G3 Mouse GE 8×60K Microarray (Agilent Technologies) for 17 hr at 65°C in a rotating Agilent hybridization oven. After hybridization, microarrays were washed for 1 min at room temperature with GE Wash Buffer 1 (Agilent Technologies), then for 1 min with GE Wash buffer 2 (Agilent Technologies) at 37°C, and dried immediately by brief centrifugation. Slides were scanned immediately on the Agilent DNA Microarray Scanner (G2505C) using one color

scan setting for 8× 60k array slides (scan area 61 × 21.6 mm, scan resolution 3 µm, dye channel was set to Green and Green PMT was set to 100%). The scanned images were analyzed with the Feature Extraction Software 10.10.1.1 (Agilent Technologies) using default parameters to obtain background subtracted and spatially detrended processed signal intensities.

## RT-PCR and qPCR

Total RNA was extracted from ES cells using the TRIzol reagent (Invitrogen) and used for cDNA synthesis with the Transcriptor First Strand cDNA Synthesis kit (Roche Applied Science, Mannheim, Germany) or PrimeScript 1st strand cDNA Synthesis Kit (Takara Bio, Otsu, Japan). All procedures were conducted according to the manufacturer's recommendations. RT-PCR was performed using KOD plus DNA polymerase (Toyobo, Osaka, Japan).

qPCR analysis was performed on a 384-well plate with QuantStudio 6 Flex Real-Time PCR System (Life Technologies) using Power SYBR® Green PCR Master Mix (Applied Biosystems, Foster City, CA). As for mRNA expression levels, the relative expression levels were normalized using *GAPDH* mRNA levels as control. The primer sequences are listed in *Supplementary file 1*.

## ChIP analysis

Cells were fixed in a medium containing 0.5% formaldehyde at room temperature for 5 min. After washing with ice-cold PBS twice, cells were resuspended in the ChIP buffer (10 mM Tris-HCl [pH 8.0], 200 mM KCl, 1 mM $CaCl_2$, 0.5% NP40) containing protease inhibitors (aprotinin, leupeptin, pepstatin A at a concentration of 1 µg/ml each) and briefly sonicated (Branson 250D sonifier, Branson Ultrasonics, Danbury, CT). After centrifugation, the supernatants containing chromatin were digested with 3 units/ml micrococcal nuclease (Worthington Biochemical, Lakewood, NJ) for 40 min at 37°C, and the reaction was stopped with ethylenediaminetetraacetic acid (EDTA; final concentration of 10 mM). Enzyme-treated supernatants were incubated with anti-mouse or anti-rabbit IgG magnetic beads (Dynabeads, Life Technologies) preincubated with anti-FLAG (M2), anti-Crm1, or anti-histone modification antibodies (2 µg) for 6 hr. (For Pol II ChIP, we first incubated the beads with rabbit anti-Rat IgG antibody [Jackson Immuno Research, West Grove, PA] before its incubation with phospho-specific Pol II antibodies.) The beads were washed thoroughly twice with each of the following three buffers, ChIP buffer, ChIP wash buffer (10 mM Tris-HCl [pH 8.0], 500 mM KCl, 1 mM $CaCl_2$, 0.5% NP40), and TE buffer (10 mM Tris-HCl [pH 8.0], 1 mM EDTA), and eluted in the elution buffer containing 50 mM Tris-HCl (pH 8.0), 10 mM EDTA, and 1% sodium dodecyl sulfate overnight at 65°C. DNA was recovered using the DNA gel extraction kit (Promega, Madison, WI) and used for ChIP-qPCR analysis or ChIP-seq analysis.

## ChIP sequencing

The ChIP library was prepared according to the Illumina protocol and sequenced on the Illumina HiSeq1500 system. The sequence reads were uniquely mapped to the reference mouse genome (mm9) using Bowtie 2 software (version 2.2.2) with default parameter (*Langmead and Salzberg, 2012*). PCR duplicates were removed from mapped reads using SAMtools (version 0.1.19).

## ChIP-seq data analysis

The mapped reads of ChIP and input DNA control data were counted in non-overlapping 200 base windows on the genome and the counts were normalized as RPM (read per million). We then calculated normalized ChIP-Seq signal intensities on each window as RPM difference between ChIP and input DNA control data ($RPM_{ChIP} - RPM_{input}$). We used MACS (version 2.0.10) with the default parameters to detect ChIP-Seq signal enriched regions.

## Accession numbers

The microarray data have been deposited in the Gene Expression Omnibus (GEO) database under the series entry GSE67967. The ChIP-Seq data are accessible through GEO Series accession number GSE68783.

## Acknowledgements

We thank Dr. Hitoshi Niwa for EB3 cells, Drs. Yoichi Miyamoto, Munehiro Asally, Tomoko Yamaguchi and Kenji Kawabata for valuable discussions, and Ms. Hitomi Inoue for technical support. This work was supported in part by Grant-in-Aid for Scientific Research on Innovative Areas (#25116008 to YY and MO) from the Ministry of Education, Culture, Sports, Science, and Technology of Japan, and Grant-in-Aids for Scientific Research (C) (#23570228 to MO) from the Japan Society for the Promotion of Science.

## Additional information

### Funding

| Funder | Grant reference number | Author |
| --- | --- | --- |
| Ministry of Education, Culture, Sports, Science, and Technology | 25116008 | Masahiro Oka<br>Yoshihiro Yoneda |
| Japan Society for the Promotion of Science | 23570228 | Masahiro Oka |

The funders had no role in study design, data collection and interpretation, or the decision to submit the work for publication.

### Author contributions

MO, Conception and design, Acquisition of data, Analysis and interpretation of data, Drafting or revising the article; SM, KY, PS, SH, KM, Acquisition of data, Analysis and interpretation of data; KK, TT, HK, Analysis and interpretation of data, Contributed unpublished essential data or reagents; YO, Analysis and interpretation of data, Drafting or revising the article; YY, Conception and design, Analysis and interpretation of data, Drafting or revising the article

### Author ORCIDs

Hiroshi Kimura, http://orcid.org/0000-0003-0854-083X

## Additional files

### Supplementary files

• Supplementary file 1. List of primers used in this study.

### Major datasets

The following datasets were generated:

| Author(s) | Year | Dataset title | Dataset ID and/or URL | Database, license, and accessibility information |
| --- | --- | --- | --- | --- |
| Oka M, Yoneda Y | 2015 | Gene expression profiling of Nup98HoxA9 expressing ES cells | http://www.ncbi.nlm.nih.gov/geo/query/acc.cgi?acc=GSE67967 | Available through the NCBI Gene Mapping Omnibus (Geo) Accession no: GSE67967 |
| Oka M, Yoneda Y, Maehara K, Hirata S, Ohkawa Y | 2015 | Genome-wide mapping of Nup98HoxA9- and Crm1-binding sites in ES cell lines. | http://www.ncbi.nlm.nih.gov/geo/query/acc.cgi?acc=GSE68783 | Available through the NCBI Gene Mapping Omnibus (Geo) Accession no: GSE68783 |

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
