## [Decision Letter]

Thank you for submitting your work entitled "Chromatin-prebound Crm1 recruits Nup98-HoxA9 fusion to induce aberrant expression of *Hox* cluster genes" for peer review at *eLife*. Your submission has been favorably evaluated by James Manley (Senior editor), Karsten Weis (Reviewing editor), and Valerie Doye (peer reviewer).

The reviewers have discussed the reviews with one another and the Reviewing editor has drafted this decision to help you prepare a revised submission.

Summary:

In this manuscript, the authors show that constitutive (over)expression of the Nup98-HoxA9 oncogenic fusion in ES cells impairs spontaneous ES cell differentiation and induces the expression of *Hox* cluster genes in ES cells. Using microarrays, ChIP-seq and ChIP, they analyze the interplay between Crm1 and Nup98-HoxA9 on their respective recruitment to *Hox* cluster regions, *Hox* gene activation, and the epigenetic status of *Hox* genes. Genome-wide studies of ChIP-seq data further indicate that FLAG-Nup98-HoxA9 and Crm1 can more broadly recruit each other on chromatin.

This is in general a well-conducted study, and the cross-talk between Crm1 and the Nup98-HoxA9 oncogenic fusion is interesting. Furthermore, the concept that association of Crm1 with *Hox* loci might be a common molecular basis for *Hox* gene dysregulation could be important. However, some conclusions, notably those linking *Hox* regulations to the bivalent state of chromatin, are not fully supported by the data. As detailed below, some additional experiments should be included and some of the conclusions should be tone down.

Essential revisions:

1) Since all functional studies on *Hox* gene expression and recruitment of Nup98-HoxA9 and Crm1 are performed in ES cells, the authors should also use stable ES cell lines to analyze the co-localization of the Nup98-HoxA9 and as control, of Nup98FG foci, with the histone variants (H3K9me2, H3K27me3, H3K9me3) which are so far performed in HeLa cells (Figure 1).

In addition, the same labeling should also be performed in the presence of TSA.

2) In the Discussion, the authors state: "Our results demonstrate that Nup98-HoxA9/Crm1 binding to *Hox* region is sensitive to TSA treatment, which results in a robust decrease of the H3K27me3 mark on *Hox* cluster genes".

The concern here is that the authors cannot exclude that the effect they observe is not the mere consequence of increased level of Nup98-HoxA9, that may lead to the dispersion of Crm1 throughout the nucleus and its subsequent removal from its endogenous targets, without any link on H3K27me3 marks.

To clarify to that point, the authors should:

a) Perform IF with Crm1 in ES cell line control or expressing FLAG-Nup98HoxA9 (that is as in Figure 4), but in the presence or absence of TSA (as in Figure 7).

b) Determine the effect of TSA on Crm1-bound genes in control ES cells, that is, in Figure 5 (or at least in Figure 5), perform the experiment with the condition parental ES + TSA; does this impair the basal level of Crm1 recruitment?

3) In the Discussion, the authors suggest that "prebound Crm1 may recruit Nup98-NSD1, Nup98-Jarid1a, and possibly other Nup98-fusions to specific *Hox* cluster genes loci through its interaction with the Nup98FG repeat, which commonly exists in all Nup98-fusions”.

This points to one set of missing experiments, i.e. does prebound Crm1 recruit Flag Nup98-FG to *Hox* cluster genes (even without subsequently inducing their activation of any developmental phenotype)?

To clarify this point, the authors should perform ChIP-qPCR experiment as in Figure 5, but with the Nup98FG cell line(s) (#13 or #20, available, as shown in Figure 2).

4) The authors should determine to which extent the oncogenic fusion is (or is not) overexpressed compared to endogenous Nup98 (indeed, unlike in these ES cells lines, the oncogenic fusions are expressed under the endogenous Nup98 promoter in leukemic cells).

Therefore, the level of Nup98-HoxA9, Nup98FG and endogenous Nup98 should be analyzed by western blot in the stable ES cell lines as compared to wt ES cells (using an anti-Nup98 N-term antibody).

5) While the Nup98-HoxA9 stable ES cell lines show an altered morphology and a decreased spontaneous differentiation, the fact that *Hox* genes expression is altered does not prove that Nup98-HoxA9 does "suppress cell differentiation in ES cells" (see impact statement). Here only a strong correlation is provided, even if this is a likely hypothesis. This should be clarified.

Minor points [abridged]:

1) The authors should comment on the fact that there is increased binding of Crm1 to the *HoxD* cluster (Figure 4) but no increase of most *HoxD* gene expression (Figure 2).

2) The authors should clarify what they mean by "formation of higher ordered structures at *Hox* gene clusters".

3) Alkaline phosphatase colonies: in Figure 3, it seems that there are more colonies in the -LIF for Nup153HoxA9 as compared to control or other constructs. We assume that this reflects variations from experiment to experiment. It would be better to provide numbers (for Figure 3/Figure 2), i.e. statistics from 3 independent plates of Alk Phosphatase positive colony number per plate or integrated AP intensity on the plate if individual colonies cannot be easily counted.

[Editors' note: further revisions were requested prior to acceptance, as described below.]

Thank you for resubmitting your work entitled "Chromatin-prebound Crm1 recruits Nup98-HoxA9 fusion to induce aberrant expression of *Hox* cluster genes" for further consideration at *eLife*. Your revised submission has been favorably evaluated by James Manley (Senior editor), Karsten Weis (Reviewing editor), and Valerie Doye (peer reviewer).

During the revision you were able to address the major issues, and in many cases, novel experiments were performed and included as main figures or supplemental figures. Therefore, the manuscript has been significantly improved but there are three remaining issues that need to be addressed before acceptance, as outlined below:

1) In the requested quantification of alkaline phosphatase positive colonies (minor point 3, "it seems that there are more colonies in the -LIF for Nup153HoxA9 as compared to control or other constructs"), the authors have indeed observed that there are more Nup153HoxA9-expressing ES cells that are AP positive. This was not what was anticipated based on the text of the previous version. As a consequence, the sentence "Furthermore, only Nup98-HoxA9 ES cells showed a significant inhibition of cell differentiation when cultured without LIF " has now simply been removed from the revised version.

In their rebuttal, the authors indicate "we speculate that this phenotype might be related to the function of Nup153 in maintaining stem cell pluripotency in ES cells, as demonstrated in a recent study (Jacinto et al., 2015)".

This point should be clearly stated in the manuscript (by including the above sentence from the rebuttal letter). In the current version, there is a supplemental figure that includes this unexpected finding, but yet without any comment.

It should be also noted that this suggests that the impact statement (while it has been toned down a bit in response to former major point 5), is still somewhat misleading. This manuscript provides merely a correlation, but not a direct demonstration, of a link between the observations of (i) Nup98-HoxA9 forming aggregates (Nup214 may sometimes form them as well), (ii) *Hox* gene regulations (only observed with Nup98-HoxA9, not with other FG-HowA9), and ES cell differentiation (impared in ES cells expressing both Nup98-HoxA9 and Nup153-HoxA9). I would suggest to further tone down the impact statement towards something like:

“Nup98-HoxA9, together with chromosomally pre-bound Crm1, a nuclear export factor, is recruited to *Hox* gene cluster regions, inducing aberrant expression of several *Hox* genes and affecting differentiation of ES cells”, or similar.

2) In response to major point 3 – "does pre-bound Crm1 recruit Flag Nup98-FG to Hox cluster genes" –, the authors have performed the suggested experiment revealing that Nup98FG is hardly accessible to genomic DNA, including the *Hox* cluster region. This is presented for the reviewers’ consideration. It would be best to add one sentence in the text of the Discussion, to clarify that (e.g. re-use the sentence from the rebuttal letter, and include Author response image 1 as a supplemental figure).

3) Unfortunately, this is a new point but it should be easy to address with the already available data. It would be very useful to have a comparison of all 3 conditions in a unique Venn diagram (654 Crm1-bound genes in control cells, 1259 Crm1-bound in Nup98-HoxA9 ES cells, and 833 FLAG-Nup98-HoxA9 bound genes). This so far appears separately in diagrams c and d. This would be very helpful as it would make it obvious how many genes are, as the *Hox* genes, bound in all 3 conditions, how many are, as *Skap2* (arrow in Figure 5), bound by Crm1 only in the presence of Nup98-HoxA9. Such a Venn diagram should be included as a main figure.

---

## [Author Response]

*Essential revisions:1) Since all functional studies on* Hox *gene expression and recruitment of Nup98-HoxA9 and Crm1 are performed in ES cells, the authors should also use stable ES cell lines to analyze the co-localization of the Nup98-HoxA9 and as control, of Nup98FG foci, with the histone variants (H3K9me2, H3K27me3, H3K9me3) which are so far performed in HeLa cells (Figure 1).*

In addition, the same labeling should also be performed in the presence of TSA.

As suggested by the reviewers, we performed immunofluorescence (IF) analysis. Our data demonstrated the partial co-localization of Nup98-HoxA9 with H3K9me2 or H3K27me3 in ES cells (Figure 2—figure supplement 1). However, since the number of Nup98-HoxA9 dots in ES cells was much lower than that in HeLa cells, it was somewhat difficult to observe the clear tendency of their association by IF. As for the effect of TSA, although the number of Nup98-HoxA9 dots drastically increased after the treatment, we also observed a significant decrease in H3K9me2 and H3K27me3 staining. Thus, their co-localization was hardly detected after TSA treatment. These results are mentioned in the text of the revised version as follows: “Immunocytochemical analysis in ES cells (Figure 2—figure supplement 1) also showed the partial co-localization of Nup98-HoxA9 dots with H3K9me and H3K27me3, but less with H3K9me3.”

2) In the Discussion, the authors state: "Our results demonstrate that Nup98-HoxA9/Crm1 binding to Hox region is sensitive to TSA treatment, which results in a robust decrease of the H3K27me3 mark on Hox cluster genes".

The concern here is that the authors cannot exclude that the effect they observe is not the mere consequence of increased level of Nup98-HoxA9, that may lead to the dispersion of Crm1 throughout the nucleus and its subsequent removal from its endogenous targets, without any link on H3K27me3marks.

To clarify to that point, the authors should:

*a) Perform IF with Crm1 in ES cell line control or expressing FLAG-Nup98HoxA9 (that is as in Figure 4), but in the presence or absence of TSA (as in Figure 7).*

b) Determine the effect of TSA on Crm1-bound genes in control ES cells, that is, in Figure 5 (or at least in Figure 5), perform the experiment with the condition parental ES + TSA; does this impair the basal level of Crm1 recruitment?

We thank the reviewers for the suggestion. Our IF data showed that, indeed, the endogenous Crm1 is released from the nuclear pore complex and scattered throughout the nucleus in the FLAG-Nup98HoxA9 ES cells in the presence of TSA, but not in its absence (Figure 7—figure supplement 1). However, our ChIP-qPCR analysis clearly demonstrated that Crm1 binding to the *Hox* cluster region is sensitive to TSA even in control ES cells (Figure 7—figure supplement 3). Collectively, we concluded that epigenetic alteration on *Hox* cluster region induced by TSA treatment, including a robust decrease of the H3K27me3 mark, is most likely the primary cause of the phenomenon. The results are described in the revised version as follows: “Furthermore, our ChIP-qPCR analysis showed that the binding of endogenous Crm1 to *Hox-A* cluster in control ES cells was also sensitive to TSA treatment (Figure 7—figure supplement 3).”

3) In the Discussion, the authors suggest that "prebound Crm1 may recruit Nup98-NSD1, Nup98-Jarid1a, and possibly other Nup98-fusions to specific Hox cluster genes loci through its interaction with the Nup98FG repeat, which commonly exists in all Nup98-fusions”.

*This points to one set of missing experiments, i.e. does prebound Crm1 recruit Flag Nup98-FG to* Hox *cluster genes (even without subsequently inducing their activation of any developmental phenotype)?*

To clarify this point, the authors should perform ChIP-qPCR experiment as in Figure 5, but with the Nup98FG cell line(s) (#13 or #20, available, as shown in Figure 2).

As suggested, we performed ChIP-qPCR analysis using the Nup98FG cell line. Our data showed that FLAG-Nup98FG could only weakly bind to the *Hox-A* cluster region. Since our IF data showed that (i) Nup98FG dots are frequently localized within the nucleolus (data not shown) and that (ii) Nup98FG is more prone to form nuclear aggregates than Nup98-HoxA9, with only a weak nucleoplasmic diffuse staining (see Figure 2), we speculate that Nup98FG is hardly accessible to genomic DNA, including the *Hox* cluster region.

4) The authors should determine to which extent the oncogenic fusion is (or is not) overexpressed compared to endogenous Nup98 (indeed, unlike in these ES cells lines, the oncogenic fusions are expressed under the endogenous Nup98 promoter in leukemic cells).

Therefore, the level of Nup98-HoxA9, Nup98FG and endogenous Nup98 should be analyzed by western blot in the stable ES cell lines as compared to wt ES cells (using an anti-Nup98 N-term antibody).

As suggested, we performed western blotting using an antibody that recognizes FG repeats of Nup98 (Figure 2—figure supplement 2). The data showed that the protein level of FLAG-Nup98-HoxA9 was relatively low compared with endogenous Nup98. Therefore, it is likely that the fusion gene was not overexpressed. We also noticed that FLAG-Nup98FG was somewhat more expressed as compared with FLAG-Nup98-HoxA9 or endogenous Nup98. These findings are described in the revised version as follows: “Immunoblotting using a monoclonal antibody that is raised against the N-terminal (A.A. 1-466) of Nup98 (Fukuhara et al., 2005) revealed that FLAG-Nup98-HoxA9 was not overexpressed compared with endogenous Nup98 (Figure 2—figure supplement 2).”

*5) While the Nup98-HoxA9 stable ES cell lines show an altered morphology and a decreased spontaneous differentiation, the fact that* Hox *genes expression is altered does not prove that Nup98-HoxA9 does "suppress cell differentiation in ES cells" (see impact statement). Here only a strong correlation is provided, even if this is a likely hypothesis. This should be clarified.*

As suggested, we modified the sentence in the impact statement as follows; “Nup98-HoxA9 is involved in the formation of the protein complex at *Hox* gene cluster regions together with chromosomally pre-bound Crm1, a nuclear export factor, to induce aberrant gene expression and affects differentiation in ES cells.”

*Minor points [abridged]:1) The authors should comment on the fact that there is increased binding of Crm1 to the* HoxD *cluster (Figure 4) but no increase of most* HoxD *gene expression (Figure 2).*

As suggested, we inserted the following comment: “It is noteworthy that, as most of the genes in the *Hox-D* cluster were not upregulated by Nup98-HoxA9, the significance of its binding to this region remains unknown.”

*2) The authors should clarify what they mean by "formation of higher ordered structures at* Hox *gene clusters".*

We apologize for the confusion. We meant, “higher order chromatin structures” instead of “higher ordered structures”. We have corrected this in the revised version.

3) Alkaline phosphatase colonies: in Figure 3, it seems that there are more colonies in the -LIF for Nup153HoxA9 as compared to control or other constructs. We assume that this reflects variations from experiment to experiment. It would be better to provide numbers (for Figure 3/Figure 2), i.e. statistics from 3 independent plates of Alk Phosphatase positive colony number per plate or integrated AP intensity on the plate if individual colonies cannot be easily counted.

We counted the number of AP-positive colonies from three independent plates and performed statistical analysis. As shown in Figure 2—figure supplement 3, our data showed that the expression of Nup98-HoxA9 conferred resistance to spontaneous differentiation of mouse ES cells in the absence of LIF. In addition, this phenotype was not observed in most of the other tested cell lines, including Nup98FG- or HoxA9-Ct-expressing ES cells. As for Nup153HoxA9-expressing ES cells, we constantly observed substantial numbers of AP-positive colonies in the absence of LIF. We speculate that this phenotype may be related to the function of Nup153 in maintaining stem cell pluripotency in ES cells, as demonstrated in a recent study (Jacinto et al., 2015).

[Editors' note: further revisions were requested prior to acceptance, as described below.]

1) In the requested quantification of alkaline phosphatase positive colonies (minor point 3, "it seems that there are more colonies in the -LIF for Nup153HoxA9 as compared to control or other constructs"), the authors have indeed observed that there are more Nup153HoxA9-expressing ES cells that are AP positive. This was not what was anticipated based on the text of the previous version. As a consequence, the sentence "Furthermore, only Nup98-HoxA9 ES cells showed a significant inhibition of cell differentiation when cultured without LIF " has now simply been removed from the revised version.

In their rebuttal, the authors indicate "we speculate that this phenotype might be related to the function of Nup153 in maintaining stem cell pluripotency in ES cells, as demonstrated in a recent study (Jacinto et al., 2015)".This point should be clearly stated in the manuscript (by including the above sentence from the rebuttal letter). In the current version, there is a supplemental figure that includes this unexpected finding, but yet without any comment.

As suggested by the reviewers, we inserted the following sentence: “Of note, we constantly observed substantial numbers of alkaline phosphatase-positive colonies of Nup153HoxA9-expressing ES cells in the absence of LIF (Figure 2 and Figure 2—figure supplement 3), which may be related to the function of Nup153 in maintaining stem cell pluripotency in ES cells, as demonstrated in a recent study (Jacinto et al., 2015).”

*It should be also noted that this suggests that the impact statement (while it has been toned down a bit in response to former major point 5), is still somewhat misleading. This manuscript provides merely a correlation, but not a direct demonstration, of a link between the observations of (i) Nup98-HoxA9 forming aggregates (Nup214 may sometimes form them as well), (ii)* Hox *gene regulations (only observed with Nup98-HoxA9, not with other FG-HowA9), and ES cell differentiation (impared in ES cells expressing both Nup98-HoxA9 and Nup153-HoxA9). I would suggest to further tone down the impact statement towards something like:*

“Nup98-HoxA9, together with chromosomally pre-bound Crm1, a nuclear export factor, is recruited to Hox gene cluster regions, inducing aberrant expression of several Hox genes and affecting differentiation of ES cells”, or similar.

As suggested by the reviewers, we modified our impact statement: “Nup98-HoxA9, together with chromosomally pre-bound Crm1, a nuclear export factor, is recruited to *Hox* gene cluster regions, inducing aberrant expression of several *Hox* genes and affecting differentiation of ES cells.”

2) In response to major point 3 – "does pre-bound Crm1 recruit Flag Nup98-FG to Hox cluster genes" –, the authors have performed the suggested experiment revealing that Nup98FG is hardly accessible to genomic DNA, including the Hox cluster region. This is presented for the reviewers’ consideration. It would be best to add one sentence in the text of the Discussion, to clarify that (e.g. re-use the sentence from the rebuttal letter, and include Author response image 1 as a supplemental figure).

As suggested by the reviewers, in the Discussion we included the data as Figure 5—figure supplement 1, and added the sentences “Unexpectedly, our data showed that Nup98FG could only weakly bind to *Hox-A* cluster region (Figure 5—figure supplement 1). However, since Nup98FG dots are frequently localized within the nucleolus (data not shown), and Nup98FG is more prone to form nuclear aggregates than Nup98-HoxA9 with only a weak nucleoplasmic diffuse staining (see Figure 2), we speculate that Nup98FG is hardly accessible to genomic DNA, including the *Hox* cluster region, when expressed by itself.”

*3) Unfortunately, this is a new point but it should be easy to address with the already available data. It would be very useful to have a comparison of all 3 conditions in a unique Venn diagram (654 Crm1-bound genes in control cells, 1259 Crm1-bound in Nup98-HoxA9 ES cells, and 833 FLAG-Nup98-HoxA9 bound genes). This so far appears separately in diagrams c and d. This would be very helpful as it would make it obvious how many genes are, as the* Hox *genes, bound in all 3 conditions, how many are, as* Skap2 (*arrow in Figure 5), bound by Crm1 only in the presence of Nup98-HoxA9. Such a Venn diagram should be included as a main figure.*

We thank the reviewers for the suggestion. We added a new Venn diagram as Figure 6.